

# Validation of Aeolus winds using ground-based radars in Antarctica and in northern Sweden

Evgenia Belova[1], Sheila Kirkwood[1], Peter Voelger[1], Sourav Chatterjee[2], Karathazhiyath Satheesan[3], Susanna Hagelin[4], Magnus Lindskog[4], and Heiner Körnich[4]

[1]Swedish Institute of Space Physics, Kiruna, SE-98128, Sweden
[2] National Centre for Polar and Ocean Research, Ministry of Earth Sciences, Goa, 403804, India
[3]Department of Atmospheric Sciences, School of Marine Sciences Cochin University of Science and Technology, Cochin, Kerala, 682 016, India
[4] Swedish Meteorological and Hydrological Institute, Norrköping, SE-60176, Sweden

Correspondence to: Evgenia Belova (evgenia.belova@irf.se)

**Abstract.**

Winds measured by lidar from the Aeolus satellite are compared with winds measured by two ground-based radars, MARA in Antarctica  (70.77° S, 11.73° E) and ESRAD (67.88° N, 21.10° E) in Arctic Sweden.  Aeolus  is a  demonstrator mission to test whether winds measured by Doppler lidar from space can have sufficient accuracy to contribute to improved weather forecasting. A comprehensive programme of calibration and validation has been undertaken following the satellite launch in 2018 but, so far, direct comparison with independent measurements from the Arctic or Antarctic regions have not been made. The comparison covers heights from the low troposphere to just above the tropopause. Results for each radar site are presented separately for Rayleigh (clear) winds, Mie (cloudy) winds, summer and winter, and ascending and descending satellite tracks. Horizontally-projected line-of-sight (HLOS) winds from Aeolus, for passes within 100 km from the radar sites, are compared with HLOS winds calculated from one-hour averaged radar horizontal wind components. The agreement in most data subsets is very good, with no evidence of significant biases (< 1 m s$^{-1}$). Possible biases are identified for two subsets, about -2 m s$^{-1}$ for MARA/Rayleigh/descending/winter winds, about 3 m s$^{-1}$ for ESRAD/Mie /ascending /winter , but these are only marginally significant. A robust significant bias of about 6 m s$^{-1}$ is found for MARA/Mie/ascending/summer winds. There is also some evidence for increased random error (by about 1 m s$^{-1}$) for all of the Aeolus winds at MARA in summer compared to winter. This might be related to the presence of sunlight scatter over the whole of Antarctica as Aeolus transits across it during summer.





# 1 Introduction

The Aeolus satellite is a European Space Agency (ESA) mission which aims to demonstrate the possibility of providing global
wind measurements throughout the troposphere and lower stratosphere using doppler wind lidar, with good enough accuracy
for use in assimilations for numerical weather prediction. Aeolus carries a single instrument - the Atmospheric Laser Doppler
Instrument (ALADIN) - with two detectors to analyse the backscattered laser light from atmospheric molecules (Rayleigh
scatter) and cloud/aerosol particles (Mie scatter), respectively (Stoffelen et al., 2005; ESA, 2008; Reitebuch, 2012). The line-
of-sight component of the wind is calculated from the Doppler shift of the backscattered light. Accurate measurement of the
Doppler shift of the backscattered light requires careful calibration of the detectors and a comprehensive process of calibration
and validation was planned for the mission. More details on the lidar performance and sources of systematic and random errors
are described in  Reitebuch et al. (2020) and Rennie and Isaksen (2020). Dedicated campaigns for comparative measurements
by airborne lidar and radiosonde profiling took place early in the mission (Baars et al., 2020, Lux et al. 2020 and Witschas et
al. 2020). Comparisons with ground-based radar and lidar observations, regular meteorological measurements and global
assimilations have also been undertaken both for specific locations (Khaykin et al. 2020, Guo et al., 2020) and on a global
scale (Martin et al., 2020, Rennie and Isaksen 2020).  Particularly the dedicated campaign and global-scale studies have led to
several changes in the data processing as the instrument performance in the space environment became better understood. For
example, corrections have had to be made for 'hot pixels' (Weiler et al., 2020) and for biases in line-of-sight winds of up to 5
m s$^{-1}$, which were found to differ between ascending and descending nodes, and between different geographic regions (Martin
et al., 2020, Rennie and Isaksen, 2020). These variable biases have been found to be largely caused by varying temperature
gradients across the instrument's mirror and, after application of corrections for the mirror effects, the biases have been reduced
considerably (to < 2 m s$^{-1}$), sufficiently for Aeolus winds to be able to improve weather forecasts (Rennie and Isaksen, 2020).
Random errors have been found to be larger than the initial goal (1-2 m s$^{-1}$) due to reduced laser power and signal losses in the
receiver path (Reitebuch et al., 2020). They have been found to be 4-5 m s$^{-1}$ (Rayleigh) or 3 m s$^{-1}$ (Mie), at least up to February
2020 (Rennie and Isaksen, 2020). So far, validation for the polar regions has been based on the ECMWF global assimilation
model (Rennie and Isaksen 2020).  Very few standard upper-air meteorological measurements (radiosondes, aircraft in-situ
sensors) are available in the polar regions so the model's accuracy is not well known in those regions.  There is a risk that
different cloud conditions, surface reflectivities and summer daylight in these regions could lead to different performance of
the lidar measurements. At the same time, accurate lidar measurements over the polar regions would be a particular asset to
global weather forecasting and climate monitoring as these regions are so poorly covered by other observations. A particular
limitation of the Aeolus dawn-dusk orbit is that measured winds are always made at the same local times. In order to provide
full coverage of the daily variation of winds, future space borne Doppler wind lidars should be deployed in orbits at other local
times but these will inevitably be more affected by the background of scattered sunlight (Zhang et al., 2020). Validation of the
Aeolus winds against direct independent wind measurements at polar latitudes offers a possibility to begin to see to what extent
polar conditions might affect measurements, particularly whether scattered sunlight effects are as theoretically predicted, since





the long summer days result in ALADIN measurements being made in full sunlight in those regions even with a dawn-dusk orbit.

The aim of this paper is therefore to use measurements from two wind-profiler radars, MARA in Antarctica and ESRAD in Arctic Sweden, to test whether the Aeolus bias corrections are good enough in those areas and to test whether there are

differences in performance between winter and summer which might be caused by scattered sunlight effects.

## 2 Overview of Measurement Platforms

### 2.1 Aeolus

The Aeolus satellite was launched on 22 August 2018 and lies in a dawn-dusk sun synchronous orbit (inclination 97°), at a height of about 320 km and with a period of about 90 minutes. The laser is pointed downwards at 35° from nadir. In most

atmospheric conditions, vertical winds are small so that the doppler shift of the backscattered light is mainly determined by the line-of-sight component of the horizontal wind. The laser is directed approximately perpendicular to the orbit track, towards the night hemisphere so as to minimise background scatter due to sunlight. During the morning (descending) part of the track the azimuth of the target-to-laser direction is about 100°, and during the evening (ascending) part the azimuth is about 260°, so that the Doppler shift of the backscattered signal is mainly due to zonal winds. At high latitudes the azimuth changes

gradually and, as a consequence, meridional winds have more effect on the Doppler shift. Measurements typically are made using 20 laser pulses (3 km of track distance), with returns integrated before height-profiles of scattering characteristics and Doppler shift are extracted. In the early part of the mission, 30 of these measurements were then analysed as a group to pick separate valid Mie (cloudy) and Rayleigh (clear) returns, leading to profiles of line-of-sight wind corresponding to about 87 km horizontal track (Reitebuch et al., 2020). The group length for Mie winds was later reduced to 14 km for better impact on

weather prediction (Rennie and Isaksen, 2020).

Processed data including the mirror correction (baseline 2B10) have been available for new observations since April 2020, however a different algorithm for the mirror correction was introduced in October 2020 (baseline 2B11). A period of homogeneous reprocessed data is also available using baseline 2B10 from July-December 2019.

### 2.2 MARA radar

The MARA wind-profiler radar is located at the Indian Antarctic station, Maitri, at 70.77° S, 11.73° E. It has been operational at that site since February 2014 (earlier located at other sites in Antarctica, see e.g. Kirkwood et al., 2007; Mihalikova and Kirkwood, 2013). It operates continuously except for occasional breaks for repairs and maintenance, and measures three components of wind in the troposphere/lowermost stratosphere at heights from about 500 m to 12 km, with interlaced measurement modes (one minute each) giving height resolution of 75 m, 150 m, and 600 m. The radar operates at a frequency

of 54.5 MHz and signal is scattered back from irregularities in the refractive index of the air. The radar beam is vertical, with a beam width of about 12°, corresponding to a 2000 m diameter of the measurement volume at 10 km height. Horizontal wind





is derived from the horizontal motion of the diffraction pattern of the scattered signal across the antenna field (Briggs, 1994), while vertical wind is directly derived from the Doppler shift of the received backscatter. The strength of the scattered signal (from refractive index gradients in the air) is strongly dependent on the gradient of potential temperature (Kirkwood et al., 2010). This often leads to a gap in coverage in the upper troposphere (approx. 6-9 km altitude) where the temperature gradient is adiabatic. This restricts the heights available for comparison with Aeolus, however the continuous operation allows comparison with any close Aeolus pass over the site (passes within 100 km occur about 4 times each week) over long periods of time. The location of MARA and the typical tracks for the 4 Aeolus orbits passing close-by each week are shown in Fig.1. The accuracy of wind measurements made by MARA has been assessed by comparison with regular radiosondes from a nearby station, over several months in 2014 (Belova et al, 2020). No significant biases (< 0.25 m s$^{-1}$) were found and random differences (standard deviation < 4 m s$^{-1}$, for 1 h averaged winds) were low enough to indicate the possibility of useful comparison with Aeolus.

## 2.3 ESRAD

The ESRAD wind-profiler radar is located at Esrange, near Kiruna in Arctic Sweden, at 67.88° N, 21.10° E. It has been in continuous operation at the same site since 1996 (Chilson et al., 1999). ESRAD operates with a vertical beam, with a beamwidth of about 4°, corresponding to 700 m diameter for the measuring volume at 10 km height. It is operated in a similar fashion to MARA, with interlaced measurement modes with height resolutions 75 m, 150 m and 900 m. Since 2019, for better comparison with Aeolus, high signal-to-noise-ratio (SNR) measurements with 900 m resolution have occupied 4 minutes of each 8-minute cycle. ESRAD operates at 52 MHz and, like MARA, the signals have low SNR in the upper troposphere. However, ESRAD has higher power (about 1.5 - 2 times the power of MARA in 2019/2020), and a much larger antenna field (about 5 times the area of MARA), allowing better height coverage, from about 500 m to 14000 m, with fewer data gaps in the upper troposphere. The accuracy of the wind measurements made by ESRAD has been assessed by comparison with 28 radiosondes launched at the same site, during the period January 2017- August 2019 and with the regional NWP model HARMONIE- AROME for the period September 2018 – May 2019 (Belova et al, 2020). These show a systematic underestimate of wind speed by about 8% in zonal wind and 25% in meridional wind at ESRAD, most likely due to non-random noise which cannot be easily removed. However, since the radiosonde and model comparisons cover the same time frame as the Aeolus mission, a correction for the underestimate can be applied before comparison with Aeolus. ESRAD-sonde random differences, after correction for the systematic underestimate (standard deviation <5 m s$^{-1}$, for 1 h averaged winds) are again low enough to indicate the possibility of useful comparison with Aeolus. The location of ESRAD and the typical measurement paths for the 3 Aeolus orbits which each week pass within 100 km of ESRAD are shown in Fig. 1.





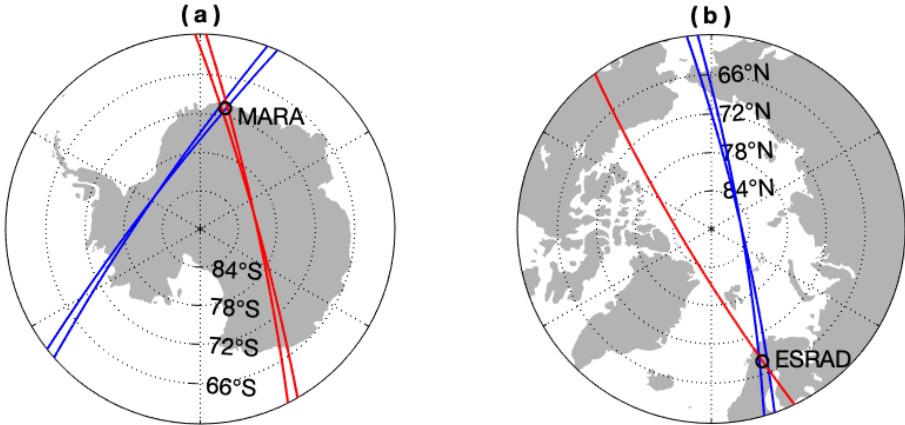

**Figure 1: Maps showing the locations of the MARA (a) and ESRAD (b) radars and typical Aeolus measurement tracks for orbits passing within 100 km of the respective radar. Red are ascending tracks, blue descending. At MARA, 4 tracks per week, at ESRAD, 3 per week, pass close to the radars.**

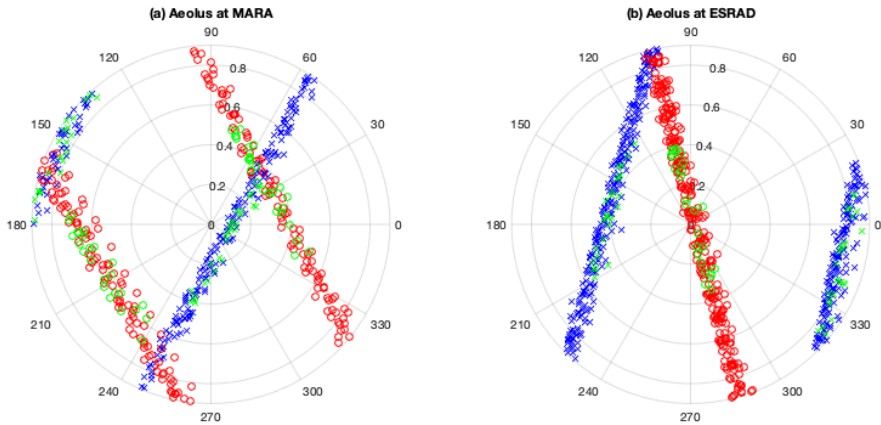

25  **Figure 2: Polar plots of azimuth and angular distance (°) from the radars of mean locations of all Aeolus wind profiles used for comparison. Mie profiles are shown in red for ascending tracks, blue descending. The along-track distance included in each Mie profile is about 14 km (0.13°). Rayleigh profiles are shown in green. The along-track distance included in each Rayleigh profile is about 87 km (0.78°).**



## 3. Description of the wind data

### 3.1 Aeolus HLOS winds

We use the Aeolus Level 2B (L2B) data product, with the 2B10 baseline which includes a bias correction for mirror temperature variations. Rayleigh clear and Mie cloudy winds are used. We use the period 1 July - 31 December 2019 where a consistent re-analysed dataset is available. Aeolus HLOS (horizontal projection of the line-of-sight component) Mie and Rayleigh winds, where the distance between the mean position of a measurement and the radar was 100 km or less, have been used for comparison. Rayleigh winds are accumulated and averaged over 87 km and we use only the measurement closest to the radar even though, sometimes, there is more than one with mean position within 100 km.  We reject Rayleigh HLOS winds with an error estimate  > 8 m s$^{-1}$ (and if the validity flag is 0). For the period July to mid October, the height coverage for Rayleigh winds over the MARA site was up to 24 or 27 km, with height resolution between 500 m and 2000 m, from mid-October and December it was up to 17 or 19 km, with height resolution between 500 m and 1000 m. For the period July to end of October, the height coverage for Rayleigh winds over the ESRAD site was up to 24 km, with height resolution between 500 m and 2000 m, from mid-October and December it was up to 20 km, with height resolution between 500 m and 1130 m (apart from a week at the beginning of November when height coverage was only up to 14 km).

Mie wind profiles are provided for 14 km accumulation lengths, so there are generally several Mie profiles within 100 km from the radar. The number of valid Mie winds in each profile is very small. Between 1 and 14 Mie wind profiles were found for each pass (mean 11), with on average only 1 valid Mie wind per profile at MARA, 2 at ESRAD. At MARA, a third of passes, at ESRAD 5% of passes, had no valid Mie winds. Valid Mie winds were found only below 11 km height at MARA, 13 km at ESRAD. Since the Mie wind data is so sparse, we average all profiles within 100 km of the radar to make a single profile, using the same height bins as the Rayleigh wind profile (closest to the radar) during the same pass. Before averaging, we reject Mie HLOS winds  with an error estimate  > 4.5 m/s (and if the validity flag is 0). The locations of  all available Aeolus wind profiles, containing at least one valid wind measurement, during the comparison period, are shown in Fig. 2.

### 3.2 MARA data

The MARA radar operates in 3 different modes, with different height coverage and resolution, interlaced, with one minute for each mode (Table 1). The length of the radar pulse determines the height resolution. Shorter pulses/smaller height resolution are aimed at the lower altitudes where the scattered signal is strong. Longer pulses, with wider height resolution are needed for the upper troposphere where the scattered signal is much weaker. The scattered signal is intrinsically highly variable in time and the 1-minute wind estimates show considerable random variation. The Aeolus measurements which we compare with correspond to an 87 km (Rayleigh), or up to 200 km (Mie) distance along the satellite track, and may be located up to 100 km from the radar, so there is no sense in comparing instantaneous measurements. We instead use averages from 30 minutes before to 30 minutes after the satellite passage. We also average to the same height intervals as the corresponding Rayleigh wind profile.





| Radar | MARA | MARA | MARA | ESRAD | ESRAD | ESRAD |
|---|---|---|---|---|---|---|
| Name | fca_75 | fcw_150 | fca_4500 | fca_150 | fca_900 | fcx_aeolus |
| Start height*, m | 150 | 200 | 5400 | 300 | 1650 | 1650 |
| End height*, m | 6200 | 13500 | 104400 | 29100 | 100650 | 27450 |
| Ht resolution, m | 75 | 150 | 600 | 150 | 900 | 900 |
| Duration, s | 60 | 60 | 60 | 120 | 120 | 120 |
| How often | 20/h | 20/h | 20/h | 8/h | 8/h | 8/h |

**Table 1: Main characteristics of the radar operating modes used in the comparison. *height above the ground. MARA (ESRAD) is located at 117 m (295 m) above the mean sea level.**

Radar wind estimates are quality checked before averaging. Checks include the absence of non-atmospheric echoes, high-enough signal-to-noise ratio (>1), mathematically successful fitting of the wind. For example, in a one-hour x 1 km (vertically) averaging bin, there are radar wind estimates from up to 60, one-minute time intervals and 20 heights at MARA. Some of these estimates will be invalid because of low signal-to-noise ratio or because the full-correlation method used to derive the wind has been unsuccessful, which particularly affects the radar measurements with shortest height resolution. In practice,

between, 100 - 400 estimates are averaged in typical height bins below 5 km altitude but only 10 - 20 above that. The large numbers of individual measurements in the lower troposphere lead to low values for the uncertainties in the mean values (standard error of the mean), typically less < 0.5 m s$^{-1}$ for MARA. In order to further reject uncertain measurements, we reject averaged winds if the standard error of the mean is > 2 m s$^{-1}$.

### 3.3 ESRAD data

The ESRAD radar operated in 4 different modes, with different height coverage and resolution, interlaced, with one or two minute for each mode. Three of these provided good enough quality data for use in this comparison (Table 1). The mode 'fcx_aeolus' was implemented, with shorter height coverage, which allows faster repetition of the radar pulses, to try to retrieve more valid wind estimates from the upper troposphere for comparison with Aeolus. As at MARA, we use averages from 30 minutes before to 30 minutes after the satellite pass and average to the same height intervals as the corresponding Rayleigh

wind profile.

As at MARA, ESRAD radar wind estimates are quality checked before averaging. At ESRAD, for example, in a one-hour x 1 km (vertically) averaging bin, there are radar wind estimates from up to 22 two-minute intervals and 8 heights at ESRAD. In practice, between 20-70 radar-wind estimates are averaged for height bins in the lower troposphere (below 6 km), 10-20 at higher altitudes at ESRAD. Uncertainties in the mean values (standard error of the mean), are typically less 1 m/s for ESRAD.

As with MARA, we reject averaged winds if the standard error of the mean is > 2 m s$^{-1}$. We also correct ESRAD wind components for the systematic underestimate found in comparison with radiosondes and reanalysis (Belova et al., 2020), 8% in zonal wind and 25% in meridional wind.



## 4. Intercomparisons

To compare radar and Aeolus winds, we first calculate what the HLOS wind should be according to the zonal (U) and meridional (V) wind components measured by the radar. (We could in principle include the vertical wind component measured by the radar, but this was found to be negligible in the one-hour averages) .

$$HLOS_{radar} = -U_{radar} \cdot sin\,(\varphi_{Aeolus}) - V_{radar} \cdot cos(\varphi_{Aeolus}) \,, \tag{1}$$

where $\varphi_{Aeolus}$ is the azimuth from the laser scattering volume to the satellite.

To quantify the differences between radar and Aeolus winds, we also compute mean differences (bias) and the standard deviation (STD) of the differences as:

$$bias = \frac{1}{N} \cdot \sum_{i=1}^{N}(HLOS_{Aeolus,i} - HLOS_{radar,i}) \tag{2}$$

$$STD = \sqrt{\frac{1}{N-1}\sum_{i=1}^{N}((HLOS_{Aeolus,i} - HLOS_{radar,i}) - bias)^{2}} \tag{3}$$

The HLOS winds measured by the radars for every Aeolus collocation event were calculated according to Equation (1) using the zonal and meridional winds averaged as described in Section 3. All data from 1 July to 31 December 2019 were divided into two seasons: summer (1 July-23 September at ESRAD, 24 September-31 December at MARA) with 12-24 hours direct sunlight and winter covering the rest of the time. Comparison between the Aeolus HLOS Rayleigh/Mie winds and HLOS winds measured by the radars has been made for each season separately. We computed correlation, a linear fit of Aeolus on the radar winds, a bias defined as the mean Aeolus-radar difference (Eq. 2), and the standard deviation (STD) of the difference (Eq. 3). In order to evaluate the uncertainties of the results, we estimated the confidence intervals for the slope of the fit and for the bias. For calculation of the altitude profiles of the bias and STD, all Aeolus and radar wind data were collected into 1-km height bins.

### 4.1 Aeolus vs MARA

In Figure 3 the comparison of Aeolus Rayleigh and MARA winds is presented. The data for the descending orbits are marked in blue and for the ascending orbits in red. We also plot there the linear fits of Aeolus on MARA winds as dashed lines. The comparison results are presented in Table 2. We see a very good agreement between the Aeolus Rayleigh and MARA HLOS winds for both seasons and pass directions: the slope of the fit is not significantly different from 1 and the bias is close to 0, with one exception - for the descending orbits in winter there is a bias of about -2 m s$^{-1}$. However, the standard deviations of the Aeolus-radar differences are relatively large (5-7 m s$^{-1}$). Since no large differences were found between the ascending and descending orbits, we made calculations for all overpasses as well.





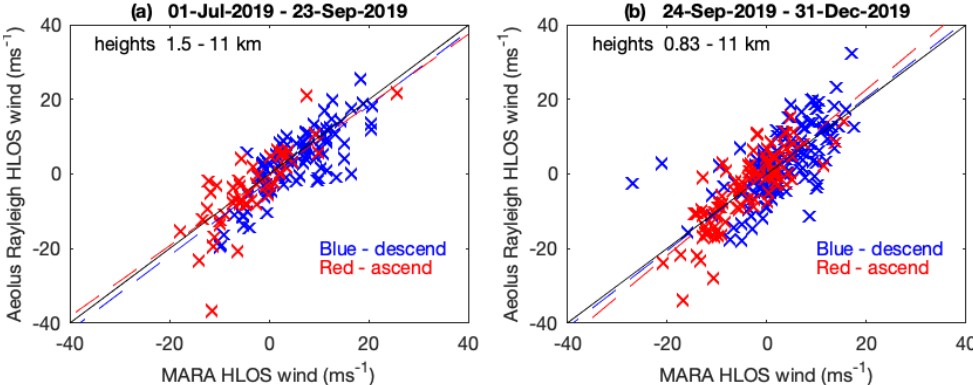

**Figure 3: Scatter plots of Aeolus Rayleigh HLOS winds against HLOS winds according to MARA radar data. Red crosses indicate measurements made on ascending tracks, blue crosses for descending. Dashed red and blue lines show fitted regression lines. Black dashed line indicates equality. Heights indicated are the lowest and highest where valid winds are available for comparison. (a) is for the Antarctic winter period 01 July-23 September 2019, (b) for summer 24 September - 31 December 2019.**

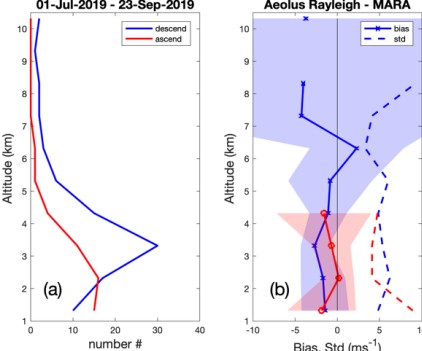

**Figure 4: Height profiles in 1-km bins of (a) the number of comparison points available and (b) mean value (bias) and standard deviation of the differences between Aeolus Rayleigh HLOS winds and MARA-derived HLOS winds for the Antarctic winter period 01 July-23 September 2019. Red lines and shading are for ascending tracks, blue for descending. Solid lines in (b) show the bias, with the shaded areas corresponding to the 90% confidence interval. Dashed lines in (b) show the standard deviation.**

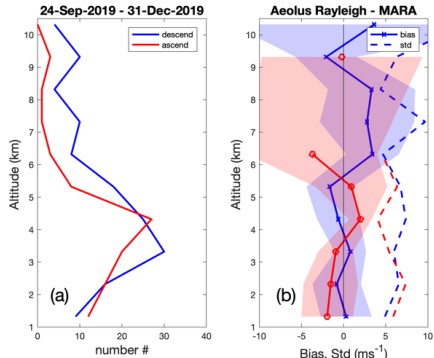

**Figure 5: the same as Fig. 4., but for Aeolus Rayleigh HLOS winds and MARA-derived HLOS winds for the Antarctic summer period 24 September - 31 December 2019.**





| Aeolus Rayleigh vs Mara | Summer 24 Sep -31 Dec 2019 | | | Winter 1 July – 23 Sep 2019 | | |
|---|---|---|---|---|---|---|
| | ascend | descend | all | ascend | descend | all |
| Altitudes, km | 0.8 – 11 km | | | 1.5 – 10.7 km | | |
| N points | 91 | 135 | 226 | 48 | 88 | 136 |
| Correlation | 0.83 | 0.66 | 0.77 | 0.80 | 0.80 | 0.82 |
| Slope, 95% conf. interval | 1.1 [0.9 1.3] | 1.0 [0.9 1.2] | 1.1 [0.9 1.1] | 0.95 [0.8 1.1] | 1.0 [0.8 1.2] | 0.95 [0.8 1.1] |
| Intercept, m s$^{-1}$ | 0.4 | 0 | 0 | -0.3 | -1.8 | -1.1 |
| Bias, m s$^{-1}$ 90% conf. interval | -0.1 [-1.1 0.9] | 0.3 [-0.7 1.3] | 0.2 [-0.6 0.9] | -0.6 [-2.1 0.9] | -1.9 [-2.9 -1.0] | -1.5 [-2.3 -0.7] |
| Std, m s$^{-1}$ | 5.7 | 7.0 | 6.5 | 6.3 | 5.4 | 5.7 |

**Table 2: Aeolus Rayleigh - MARA HLOS wind comparison.**

The behaviour of the bias and standard deviation of the Aeolus-radar differences as a function of height is shown in Figs. 4 and 5. The bias and standard deviation of the differences do not vary significantly with height. The bias uncertainties estimated at 90% confidence are reasonably small up to about 6 km altitude where there are relatively many valid data points for comparison. In Fig. 4 we can also see that the small ( -2 m s$^{-1}$) bias for winter descending orbits is not systematically significant over an extended height range.

Figures 6-8 and Table 3 show the results of the comparison for Aeolus Mie winds. The height coverage and the number of valid Mie data points is small, especially for the winter season, which leads to high uncertainties. The biases are small except for the ascending passes in summer where the bias is substantial, 6 m s$^{-1}$. This deviation is clearly seen in Fig. 6 and in the height-resolved plot in Fig. 8, between 2.5 km and 4.5 km, it is systematically significantly well above zero. We note that the bias only appears for the ascending track, and only for Mie winds, not Rayleigh winds. Closer examination of the data (not shown here) also shows that the bias affects both the closer tracks to the north-east of the radar and those to the south-west (see Fig. 2). Although not shown here, we have also compared the same Aeolus-Mie wind estimates with HLOS winds calculated from the ECMWF reanalysis (ERA5) for the MARA location. That comparison shows the Aeolus Mie winds for summer ascending tracks are, on average, 7 m s$^{-1}$ higher than the ERA5 winds.

The random differences between MARA and Aeolus HLOS winds (STD) are 5.7-7.0 m s$^{-1}$ for summer, 4.9 – 6.5 m s$^{-1}$ for winter and essentially the same for Mie and Rayleigh winds. This is much bigger than the standard error for the average radar winds themselves (2 m s$^{-1}$) and somewhat more than found comparing MARA and radiosonde winds (4 m s$^{-1}$, Belova et al, 2020). Some of this will be due to the expected random errors of the Aeolus winds (4-5 m s$^{-1}$ for Rayleigh winds, 3 m s$^{-1}$ for Mie winds) and the difference in location of MARA and Aeolus measurements (see Fig 2).

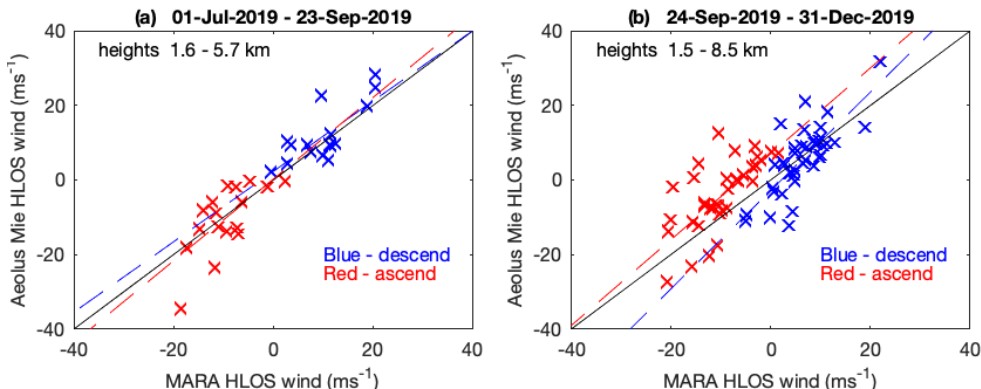

**Figure 6: the same as Fig. 3 but for Aeolus Mie HLOS winds against MARA.**

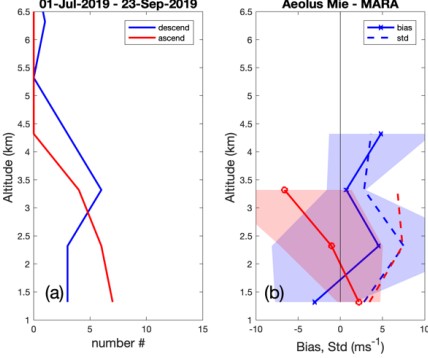

'55

**Figure 7: The same as Fig. 4 but for Aeolus Mie HLOS winds against MARA (winter).**

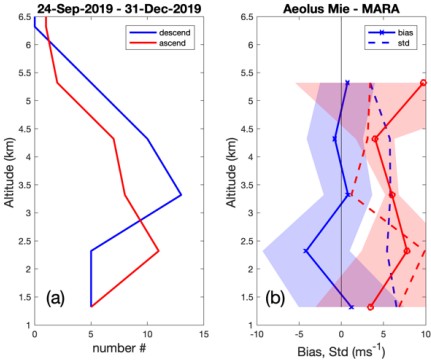

**Figure 8: The same as Fig. 5 but for Aeolus Mie HLOS winds against MARA (summer).**





| Aeolus Mie vs MARA | Summer | | Winter | |
| | 24 Sep - 31 Dec 2019 | | 1 July – 23 Sep 2019 | |
| | ascend | descend | ascend | descend |
| --- | --- | --- | --- | --- |
| Altitudes, km | 1.5 – 8.5 | | 1.6 – 5.7 | |
| N points | 35 | 39 | 17 | 16 |
| correlation | 0.69 | 0.77 | 0.71 | 0.77 |
| Slope, 95% conf. interval | 1.2 [0.8 1.5] | 1.3 [1 1.7] | 1.1 [0.4 1.8] | 0.94 [0.5 1.4] |
| Intercept, m s$^{-1}$ | 7.0 | -2.8 | 0.2 | 2.3 |
| Bias, m s$^{-1}$ 90% conf. interval | 6.0 [4.0 8.0] | -0.6 [-2.3 1.0] | -1.0 [-3.8 1.7] | 1.9 [-0.2 4.1] |
| Std, m s$^{-1}$ | 7.0 | 6.0 | 6.5 | 4.9 |

**Table 3: Aeolus Mie - MARA HLOS wind comparison.**

The distance between the observations can be up to 100 km, with the largest spread of locations for the Mie measurements, and is more than in most cases in a comparison with radiosondes. However, the systematically higher random differences in summer compared to winter suggest higher Aeolus random errors in summer, since the distances from MARA do not vary between the seasons and weather systems are more variable, which would lead to more spatial difference, in winter rather than in summer.

**4.2 Aeolus vs ESRAD**

The results of the comparison between Aeolus and ESRAD are presented in Figures 9-14 and Tables 4 and 5. In general, there are significantly more valid data points for Rayleigh, as well for Mie winds, than in comparison with MARA and height coverage is also extended. The results for Rayleigh winds are summarised in Table 4. The slopes of the linear fits are about 1, but while the biases are 0, within the uncertainties, for the ascending tracks, they are slightly negative (about -1 m s$^{-1}$ ) for the descending tracks. Although, given the rather broad confidence intervals, it is possible that a negative bias is present for both ascending and descending tracks. Again, since there are no large differences in bias or slope between ascent and descent, we also calculate for both sets together, and the results show similar negative bias to the descending tracks (which contribute more points). The height profile of the bias in Fig. 11 shows that only heights above 6 km (for winter, descending tracks), demonstrate the -1 m s$^{-1}$ bias systematically. These are heights and times where wind speeds are high and where a possible imprecise correction for the ESRAD wind-speed underestimate could have a significant effect. As a result, we cannot be sure that there is any real bias in the Aeolus Rayleigh winds.



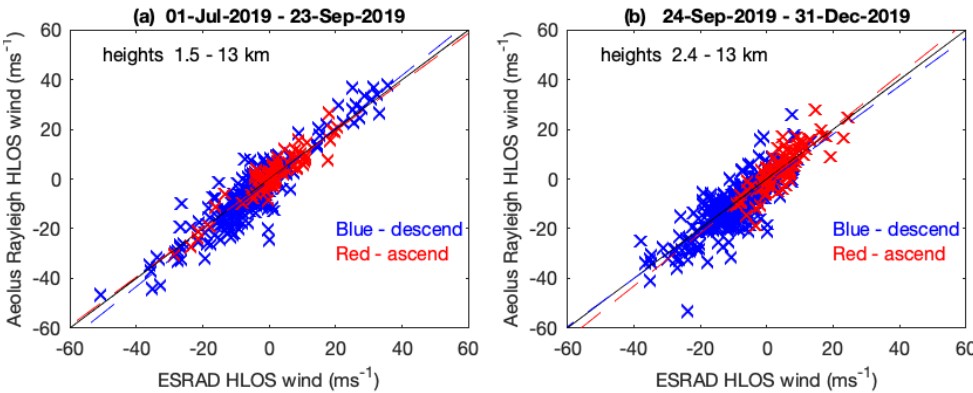

**Figure 9: Scatter plots of Aeolus Rayleigh HLOS winds against HLOS winds according to ESRAD radar data. Red crosses indicate**
**measurements made on ascending tracks, blue crosses for descending. Dashed red and blue lines show fitted regression lines. Black**
**dashed line indicates equality. Heights indicated are the lowest and highest where valid winds are available for comparison. (a) is**
**for the Arctic summer period 01 July-23 September 2019, (b) for winter 24 September - 31 December 2019.**

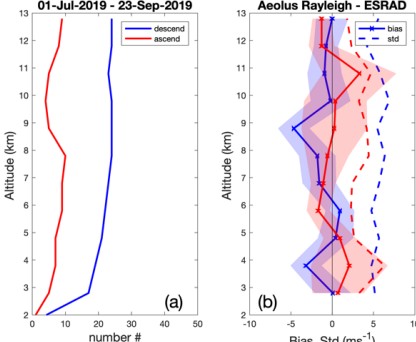

**Figure 10: Height profiles in 1-km bins of (a) the number of comparison points available and (b) mean value (bias) and standard**
**deviation of the differences between Aeolus Rayleigh HLOS winds and ESRAD-derived HLOS winds for the Arctic summer period**
**01 July-23 September 2019. Red lines and shading are for ascending tracks, blue for descending. Solid lines in (b) show the bias,**
**with the shaded areas corresponding to the 90% confidence interval. Dashed lines in (b) show the standard deviation.**

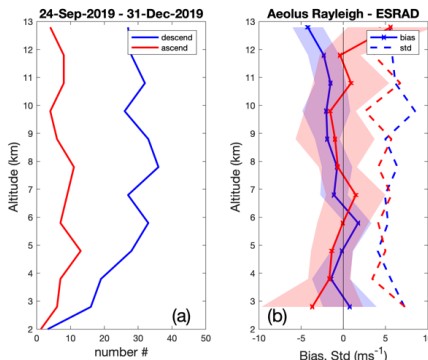

**Figure 11: The same as Fig. 10., but for Aeolus Rayleigh HLOS winds and ESRAD-derived HLOS winds for the Arctic winter period**
**24 September - 31 December 2019.**





| Aeolus Rayleigh vs ESRAD | Summer 1 July – 23 Sep 2019 | | | Winter 24 Sep – 31 Dec 2019 | | |
|---|---|---|---|---|---|---|
| | ascend | descend | all | ascend | descend | all |
| Altitudes, km | 1.5 – 12.9 | | | 2.4 – 13.3 | | |
| N points | 78 | 246 | 324 | 83 | 305 | 388 |
| Correlation | 0.93 | 0.92 | 0.92 | 0.83 | 0.83 | 0.87 |
| Slope, 95% conf. interval | 0.98 [0.89 1.1] | 1.1 [1 1.1] | 1.1 [1 1.1] | 1.1 [0.9 1.2] | 0.97 [0.9 1] | 1.0 [0.96 1.1] |
| Intercept, m s$^{-1}$ | -0.2 | -0.8 | -0.7 | -0.6 | -1.3 | -0.9 |
| Bias, m s$^{-1}$ 90% conf. interval | -0.1 [-0.8 0.6] | -1.1 [-1.7 - 0.5] | -0.9 [-1.4 -0.3] | -0.4 [-1.3 0.6] | -1.2 [-1.8 -0.6] | -1.0 [-1.6 -0.4] |
| Std, m s$^{-1}$ | 3.6 | 5.7 | 5.3 | 5.1 | 6.8 | 5.8 |

Table 4: Aeolus Rayleigh - ESRAD HLOS wind comparison.

| Aeolus Mie vs ESRAD | Summer 1 July – 23 Sep 2019 | | Winter 24 Sep – 31 Dec 2019 | |
|---|---|---|---|---|
| | ascend | descend | ascend | descend |
| Altitudes, km | 1.4 -10.9 | | 1.8 – 11.4 | |
| N points | 36 | 93 | 35 | 83 |
| Correlation | 0.83 | 0.89 | 0.82 | 0.88 |
| Slope, 95% conf. interval | 0.81 [0.68 0.95] | 0.87 [0.79 0.95] | 0.89 [0.76 1] | 1.0 [0.93 1.1] |
| Intercept | 0.6 | 0.5 | 2.4 | 0.6 |
| Bias, m s$^{-1}$ 90% conf. interval | 0.2 [-0.9 1.4] | 0.4 [-0.6 1.5] | 3.5 [1.6 5.5] | -0.4 [-1.6 0.7] |
| Std, m s$^{-1}$ | 4.0 | 6.1 | 6.8 | 6.3 |

Table 5: Aeolus Mie - ESRAD HLOS wind comparison.

For Mie winds (Figs. 12-14, Table 5), the number of available comparisons is small, but higher than at MARA and the height coverage is better. The slopes of the regression lines are close to one and the bias not significantly different from zero, except in the case of the ascending orbits in winter when the average bias is found to be 3.5 m s$^{-1}$. In Fig. 14 we can see that this bias is systematically positive at all heights, but the number of data points is very small. Further, although not shown here, this bias does not appear when we compare Aeolus winds with ECMWF reanalysis (ERA5) winds at the ESRAD location so it cannot be taken as proven.



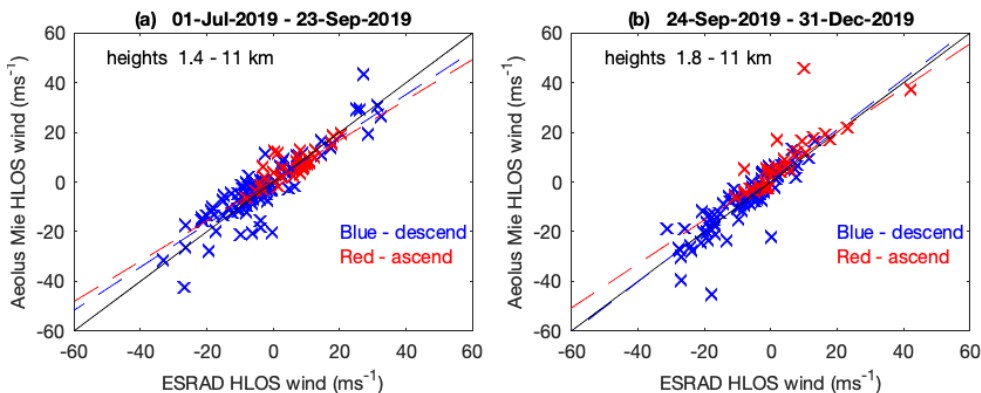

00

**Figure 12: The same as Fig. 9 but for Aeolus Mie HLOS winds against ESRAD.**

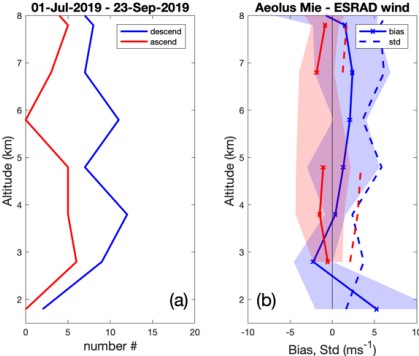

**Figure 13: As Fig. 10 but for Aeolus Mie HLOS winds against ESRAD (summer).**

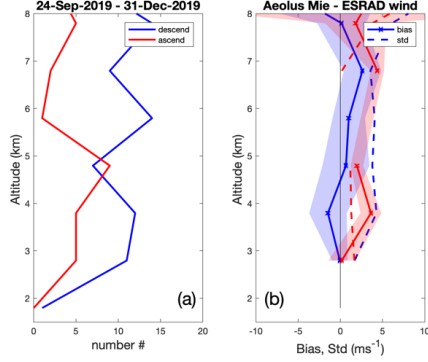

05    **Figure 14: As Fig. 11 but for Aeolus Mie HLOS winds against ESRAD (winter).**



The random differences between ESRAD and Aeolus HLOS winds (STD) are 3.6-6.1 m s$^{-1}$ for summer, 5.1-6.8 m s$^{-1}$ for winter and essentially the same for Mie and Rayleigh winds. This is again much bigger than the standard error for the average radar winds themselves (2 m s$^{-1}$) but, for summer, close to that found comparing ESRAD and radiosonde winds (4 m s$^{-1}$ , Belova et al., 2020). For winter the random differences are higher than in the comparison with radiosondes. Some of this will be due to the expected random errors of the Aeolus winds together with the distance up to 100 km between the observations. The random differences are indeed slightly smaller for the ascending paths than the descending ones and, as shown in Fig 2, the Aeolus measurements are closer to the radar site on the ascending conjunctions. The systematically higher random differences in winter compared to summer can be due to weather systems being more variable in winter than in summer.

## 5. Summary and conclusion.

The aim of this study was to assess the accuracy of winds measured by lidar from the Aeolus satellite at polar latitudes by comparison with independent measurements made by two ground-based wind-profiler radars, MARA near the coast of Queen Maud Land in Antarctica and ESRAD, in Arctic Sweden. The radars make their observations at fixed locations (within a radius of a few 100 m) hence, exactly co-located measurements with Aeolus are not possible. Aeolus Rayleigh (clear sky) measurements are averaged along a considerable accumulation length (87 km) and, although Aeolus Mie (cloudy) measurements are in principle averaged over shorter distances (14 km), the latter are in practice found to be sparse at the radar locations since they depend on the presence of suitable clouds or aerosols. For comparison with the radar we therefore use the Rayleigh (clear) wind profile closest to the radar for any pass within 100 km, and average all Mie (cloudy) winds registered within 100 km of the radar on that pass. We use temporal averaging from 30 minutes before to 30 minutes after the satellite pass for the radar winds which may provide, to some extent, a proxy for the spatial averaging intrinsic to the Aeolus measurements. We separated the datasets at each radar site into ascending and descending passes and into summer and winter seasons as they might show different behaviour. The agreement between Aeolus and radar winds is generally very good. The slopes of Aeolus-on-radar wind regression lines do not differ significantly from 1, and correlation coefficients are between 0.66 and 0.93. The random differences (4 - 7 m s$^{-1}$) are in most cases about what could be expected from the known level of random error for Aeolus winds (4-5 m s$^{-1}$ for Rayleigh, 3 m s$^{-1}$ for Mie) and spatial / temporal differences between radar and Aeolus measurements.

There is a particular interest in possible biases since these have been found to be a problem in the earlier stages of Aeolus validation, but have been much reduced by more recent data processing methods (including the 2B10 baseline which is used here). The only clear systematic bias found in this comparison is 6.0 [4.0 8.0] m s$^{-1}$ for Mie (cloudy) winds at MARA for ascending passes in summer (where the values inside the square brackets are the 90% confidence limits). This bias also appears when the Aeolus Mie winds are compared with ECMWF ERA5 reanalysis. Summer is the season when the effect of scattered sunlight from the ice-cap is maximum, and can affect the satellite for some minutes as it crosses Antarctica before passing the



MARA site. There is also some indication that the random errors of both Rayleigh and Mie winds may be about 1 m/s more in summer than winter at MARA.

There are further small biases with marginal significance. At MARA, there is a small bias -1.9 [-2.9 -1.0] m s$^{-1}$ for Rayleigh / descending / winter  but this is not systematically significant over an extended height range. At ESRAD, there appears to be a larger bias, 3.5 [1.6 5.5] m s$^{-1}$ for Mie /ascending /winter but this is based on a very small number of comparison points. At ESRAD, there appears to be a small negative bias for all  Rayleigh wind comparisons, on average -1 m s$^{-1}$ but we cannot rule out the possibility that this is a performance problem with the radar.

In summary, the agreement between radar and Aeolus winds is generally very good. For 13  out of 16 subdivisions of the data (Rayleigh/Mie, ascending/descending tracks, summer/winter, Arctic/Antarctic) we find no evidence for any bias in the Aeolus winds (< 1m s$^{-1}$).  For a further two subdivisions (specified in the previous paragraph)  there may be a significant bias but more data will be needed to establish whether this is truly the case.  We find robust evidence for a large bias  of about 6 m s$^{-1}$ in only one case - summer, Mie winds for the ascending tracks over MARA in Antarctica.  This should be looked into further when
more recent data becomes available from MARA.

**Data availability**

ESRAD data are available from PV on motivated request. MARA data can be obtained on reasonable request from SC. Aeolus data are publicly available at the ESA Aeolus Online Dissemination System.

**Author contribution**

SK, EB and PV develop and maintain the software and data processing for ESRAD. They developed the codes for the radar-Aeolus comparison and conducted the data analysis. SC and KS provided the MARA data. All co-authors discussed the comparison methods and results.  EB and SK prepared the manuscript with contribution from all co-authors.

**Competing interests**

The authors declare that they have no conflict of interest.

**Acknowledgements**

This work was supported by Swedish National Space Agency (grant numbers 125/18, 279/18). ESRAD operation and maintenance is provided by Esrange Space Center of Swedish Space Corporation. The team members at Maitri station for the 38th Indian scientific expedition to Antarctica (ISEA) and the Antarctic logistics division at NCPOR (India) are acknowledged



for providing necessary supports for the operation of MARA. Figure 1 is plotting using "M_Map: A mapping package for
MATLAB", version 1.4m, by R. Pawlowicz, 2020, available online at www.eoas.ubc.ca/~rich/map.html.

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
