# Peer review of "Validation of Aeolus winds using ground-based radars in Antarctica"

_Atmospheric Measurement Techniques, 2021_

## Author Comment (AC1)

Title: Validation of Aeolus winds using ground-based radars in Antarctica and in northern Sweden Author(s): Evgenia Belova et al. MS No.: amt-2021-54 MS type: Research article Special Issue: Aeolus data and their application (AMT/ACP/WCD inter-journal SI)

We thank the reviewer for the comments that help us to correct and improve our paper. The referee comments are in **black**, our reply is in blue and changes in the manuscript are in **magenta**.

The main addition we will make to the paper following the reviewers comments (particularly reviewer 3) is to include two new figures summarising the mean differences between Aeolus and radar winds, also showing a comparison with the ERA5 model.

In preparing these figures we realised that one of the quality checks for the radar wind data (the requirement that 95% confidence limit for the time/height average should be < 2 m/s) was not applied correctly. Correcting this leads to somewhat fewer comparison points (about 23% less for Rayleigh winds, about 13% for Mie winds) and to changes in the exact numbers for intercepts/biases/standard deviation etc in the Tables. Standard deviations are generally slightly less, biases changed by less than 1 m/s and the changes are within the confidence limits given in the original tables. Corresponding changes will be made in the text.

**Anonymous Referee #1**

The main analysis recommendation is to see if more insight can be gained by digging further into the origin of the differences seen between ESRAD and Aeolus, in particular to what extent biases in ESRA might account for those differences (versus the lack of coincident data, particularly in the Mie channel, which is clearly described). Regarding ESRAD biases, taking a quick look at Belova et al. (2020) AMTD (https://amt.copernicus.org/preprints/amt-2020-405/amt-2020-405.pdf), the differences of ESRAD from radiosonde, HARMONIE-AROME, and ERA5 are complex, and I couldn't find a clear statement in it about recommended bias correction to ESRAD (apologies if I missed it). I was looking for this because the nature of and rationale for the bias correction made to ESRAD in the current paper (e.g. p.7, line 85-87) is not entirely clear. At p.4, ll.14-16, what does it mean to say that there is a 'systematic underestimate of wind speed by about 8% in zonal wind and 25% in meridional wind'. Wind speed is a scalar, so is the issue that ESRAD winds are weaker (lower speed) than the other products? From the other Belova et al. (2020) paper in AMTD, it seems like these statistics refer to separate linear regressions carried out for U and V. Is this suitable to apply when comparing ESRAD to an HLOS product like Aeolus? Again, why separately correct zonal and meridional winds rather than wind speed and direction? Have the authors done a separate analysis of the comparison of ESRAD to Aeolus without the Belova et al. 2020 bias corrections on ESRAD? If so, what does this reveal?

In the published paper by Belova et al. (2021) which is revised version of Belova et al. (2020) we added discussion and explanation why ESRAD underestimates winds and why the underestimates are different for the zonal and meridional components. (Note the underestimates are by a % of the magnitude of each component, they are not a 'bias' which would imply fixed offsets). In Belova et al., 2021 we answered most of your questions. However, we will add more explanation how and why ESRAD winds were corrected for the underestimates also in the present paper:

To be added at the end of section 3.3:

Radar winds are measured from time delays between signals received on different sections of the radar antenna array as the diffraction pattern of the scattered radio waves is advected by the wind. The baseline for determining the zonal component is longer than that for the meridional one and the receivers for the different parts of the array are not equally susceptible to non-random noise. This leads to underestimates of the wind speed which differ between the two components (Belova et al., 2021)

Belova, E., Voelger, P., Kirkwood, S., Hagelin, S., Lindskog, M., Körnich, H., Chatterjee, S., and Satheesan, K.: Validation of wind measurements of two mesosphere–stratosphere–troposphere radars in northern Sweden and in Antarctica, Atmos. Meas. Tech., 14, 2813–2825, https://doi.org/10.5194/amt-14-2813-2021, 2021.

Also regarding ESRAD, it is interesting that the fcx\_aeolus was implemented as a special effort for the calibration/validation effort (p.7,1.78, and Table 1). But apart from the

mention on p.7, a separate analysis of this data does not appear. Were systematic differences were found for this mode?

The mode fcx\_aeolus was not analysed separately. It was simply designed to provide higher signal-to-noise ratio than fca\_900 in the troposphere at the expense of height coverage of the mesosphere, so as to provide more wind estimates in parts of the atmosphere where the signal is weak (e.g. upper troposphere).

The main textual revisions I recommend are to clarify in the abstract and elsewhere that the analysis is based on a single six-month period of 1 July-31 December 2019, to clarify what the nature of the 'winter' and 'summer' seasons are here, and to discuss the implications of the use of a single season to characterize these errors. The reader might assume from the abstract that the analysis would take place over the entire Aeolus period instead of just when the homogenized and reprocessed 2B10 data was available. The authors could expand on their justification of only including this data in their analysis at p.3, line 84.

We will add the dates to the first sentence in the abstract to say: Winds measured by lidar from the Aeolus satellite are compared with winds measured by two ground-based radars, MARA in Antarctica (70.77° S, 11.73° E) and ESRAD (67.88° N, 21.10° E) in Arctic Sweden, for the period 1 July - 31 December 2019.

We will add more justification at the end of section 2.1 (p3, line 84):

The move to baseline 2B10 and higher has been found to make considerable improvements to biases generally (Martin et al., 2020, Rennie and Isaksen, 2020) so it is most relevant to compare with these baselines. Because of long lead times for data transfer from Antarctica, at the time of writing, the most recent data available from MARA was 31 December 2019, so we focus on the time interval 1 July - 31 December 2019.

We will replace summer and winter by sunlit ('summer') and non-sunlit ('winter') when they are first mentioned in the abstract and the text (section 4)

Abstract :

Results for each radar site are presented separately for Rayleigh (clear) winds, Mie (cloudy) winds, sunlit ('summer') and non-sunlit ('winter'),

Section 4 (p.8, 11.04-05):

All data from 1 July to 31 December 2019 were divided into two seasons: sunlit ('summer', 1 July-23 September at ESRAD, 24 September-31 December at MARA) with 12-24 hours direct sunlight and non-sunlit ('winter') covering the rest of the time.

In addition, while it is ok to characterize 1 July- 24 September (24 September - 31 December) as boreal (austral) 'summer', the complementary periods are shoulder seasons (boreal autumn/austral spring) and not 'winter'. This nomenclature is used when the authors interpret some of the wind biases in terms of winter-versus-summer seasonality (p.12, ll.63-65; p.16, ll13-14). This interpretation should mention that results could be influenced by a small number of weather systems that happened to occur at these sites during the six-month analysis period.

It is clear that the amount of sunlight distinguishes the two periods (p.8, 11.04-05) and the reason to separate the periods in this way is to focus on the role of insolation backscatter in controlling errors. So could the authors call the 'winter' period something like the 'nonsummer' or 'non-sunlit' period?

**p.8, 11.04-05) see above**

**p.16, ll13-14) see below**

p.12, ll.63-65; We will add a sentence on the limitations of the short time interval.

However, given the very short time interval which we have analysed, it is possible that this is not a summer/winter effect but just a result of a small number of individual weather systems.

Specific comments: \* p.2, 1.43: Clarify what is meant by 'hot pixels'.

Hot pixels are increased dark current rates for specific ALADIN ACCD detector pixels, which can cause large biases in HLOS winds if not corrected for. We will add to the text:

"...corrections have had to be made for 'hot pixels 'which are increased dark current rates for specific ALADIN ACCD (accumulation charge coupled device) detector pixels..."

\* p.2, l.55: Is this comment necessary for this paper? Perhaps it would be better placed in the discussion. The description of the 'limitation' of the Aeolus orbit design is distracting. The sun-synchronous orbit presents a challenge for calibration/validation but it is a reasonable strategy for capturing free tropospheric/lower stratospheric winds whose diurnal cycle is relatively weak, especially on the typical horizontal measurement scales of 10-100km achievable by this technology.

We will remove the sentence 'A particular limitation of the Aeolus dawn-dusk orbit is that measured winds are always made at the same local times. '

\* p.3, l.78: to about \*an\* 87 km horizontal

Ok.

\* p.3, 1.79: for better impact on weather prediction -> to improve the impact of the retrieved Aeolus winds on numerical weather prediction

Ok.

\* p.6, l.38: ">8 m s^-1": While it seems reasonable, how was this rejection threshold chosen, and what impact did it have on the results?

The rejection thresholds of 4.5 m/s for Mie and 8 m/s Rayleigh errors were recommended by ESA/DISC (Aeolus Data Science and Innovation Cluster) (Reitebuch et al., 2019; Stoffelen et al., 2019; Rennie and Isaksen, 2020) for the early period of Aeolus observations (laser FM-A) for the Aeolus CAL/VAL teams. For the later period (laser FM-B) the limit for Mie was slightly changed for 5 m/s. These thresholds are chosen subjectively based on the compromise between the number of observations that pass quality control and the overall quality of the data set (Rennie and Isaksen, 2020). We followed 1st recommendation in order to get more Mie wind data for comparison and did not try other QCs. In revising the paper we will change to 5 m/s for Mie winds since this is in better agreement with the more recent recommendation. We will add to the text ( P 6 1 150, section 3.1):

The rejection thresholds for Mie and Rayleigh winds were chosen subjectively based on the compromise between the number of observations that pass quality control and the overall quality of the data set (Rennie and Isaksen, 2020).

Rennie, M. P. and Isaksen, L.: The NWP impact of Aeolus Level 2B winds at ECMWF, ECMWF technical memo, 864, doi: 10.21957/alift7mhr, 2020.

\* p.6, starting 1.45: Replace "Mie winds" with "Mie wind measurements"

Ok.

\* p.7, Table 1: Is there a typo in the end heights (104km, 100 km) - and if not is this consistent with the descriptions in Section 2?

This is correct for the end heights. These experimental modes are designed to observe the returns from mesospheric heights e.g. polar mesosphere summer echoes. In section 2 we described the lower atmosphere measurements which are relevant for Aeolus.

\* Figures 3 and 6 show strong negative values for ascending HLOS in Aeolus but not in MARA; can the authors comment on these outliers? These extreme differences might be

**worth pointing out. Is it possible that there are ranges of horizontal wind speeds that aren't captured by MARA's retrieval?**

There is no reason why MARA cannot measure strong winds - the comparison with radiosondes in Belova et al (2021) shows good agreement out to at least 40 m/s, with discrepancies in both directions (MARA wind > sonde wind, and sonde wind > MARA wind). However, given that ground level varies from close to sea level at MARA to around 2000 m altitude just 100 km to the south, and strong localised katabatic winds occur in some conditions in Antarctica, strong local variations in low-level winds could be expected. There are indications of this in the very high standard deviation at the lowest heights (Fig 4b and 8b). But there are not really enough points to make a detailed study of this.

\* p.12, 1.63: The wording is confusing here, suggest "... do not vary between the seasons and weater systems are more variable in winter rather than in summer, which would lead to more spatial variability in winter than in summer". But again, as pointed out above, it is speculative to make this kind of generalization when only analyzing a single season, especially since there is a lot of synoptic variability in Antarctica year-round.

**See above.**

\* p.16, l.10: "For winter the random differences are higher than in the comparison with radiosondes." Could this be quantified?

Since we corrected our quality check for ESRAD data, the random differences are no longer different between summer and winter. We will remove the text relating to summer winter differences.

---

## Author Comment (AC2)

Title: Validation of Aeolus winds using ground-based radars in Antarctica and in northern Sweden Author(s): Evgenia Belova et al. MS No.: amt-2021-54 MS type: Research article Special Issue: Aeolus data and their application (AMT/ACP/WCD inter-journal SI)

We thank the referee for the comments that help us to correct and improve our paper. The referee comments are in **black**, our reply is in blue and changes in the manuscript are in **magenta**.

The main addition we will make to the paper following the reviewers comments (particularly reviewer 3) is to include two new figures summarising the mean differences between Aeolus and radar winds, also showing a comparison with the ERA5 model (Figures X and Y included at the end of this reply).

In preparing these figures we realised that one of the quality checks for the radar wind data (the requirement that 95% confidence limit for the time/height average should be

Figure X. Month-by-month mean values of biases in HLOS winds (solid lines) and 90% confidence limits (shaded areas) at MARA. Red for ascending tracks, blue for descending. a) Aeolus Rayleigh minus MARA, b) Aeolus Mie minus MARA, c) Aeolus Rayleigh minus ERA5, d) Aeolus Mie minus ERA5, e) MARA minus ERA5 at available Aeolus Rayleigh comparison times/heights, f) MARA minus ERA5 at available Aeolus Mie comparison times/heights.

Figure Y: Month-by-month mean values of biases in HLOS winds (solid lines) and 90% confidence limits (shaded areas) at ESRAD. Red for ascending tracks, blue for descending. a) Aeolus Rayleigh minus ESRAD, b) Aeolus Mie minus ESRAD, c) Aeolus Rayleigh minus ERA5, d) Aeolus Mie minus ERA5, e) ESRAD minus ERA5 at available Aeolus Rayleigh comparison times/heights, f) ESRAD minus ERA5 at available Aeolus Mie comparison times/heights.

---

## Author Comment (AC3)

Title: Validation of Aeolus winds using ground-based radars in Antarctica
and in northern Sweden
Author(s): Evgenia Belova et al.
MS No.: amt-2021-54
MS type: Research article
Special Issue: Aeolus data and their application (AMT/ACP/WCD inter-journal SI)

We thank the referee for the comments that help us to correct and improve our paper.
The referee comments are in black, our reply is in blue and changes in the manuscript are in magenta.

The main addition we will make to the paper following the reviewers comments (particularly reviewer 3) is to include two new figures summarising the mean differences between Aeolus and radar winds, also showing a comparison with the ERA5 model.
In preparing these figures we realised that one of the quality checks for the radar wind data (the requirement that 95% confidence limit for the time/height average should be < 2 m/s) was not applied correctly. Correcting this leads to somewhat fewer comparison points (about 23% less for Rayleigh winds, about 13% for Mie winds) and to changes in the exact numbers for intercepts/biases/standard deviation etc in the Tables. Standard deviations are generally slightly less, biases changed by less than 1 m/s and the changes are within the confidence limits given in the original tables. Corresponding changes will be made in the text.

Anonymous Referee #3

General comments
In the manuscript the reprocessed data set from 1 July 2019 until 31 December 2019 is validated. First of all, this is an important information which should be mentioned in the abstract.

We will add the dates to the first sentence in the abstract to say:

Winds measured by lidar from the Aeolus satellite are compared with winds measured by two ground-based radars, MARA in Antarctica (70.77° S, 11.73° E) and ESRAD (67.88° N, 21.10° E) in Arctic Sweden, for the period 1 July - 31 December 2019.

The second point is, did you also analyze the operational data set from this time period in order to estimate the improvements of the new processor versions especially at the locations near the poles, where this information could be of interest for the processor developers?

We did not really look at the operational datasets (baselines 2B06/2B07) since our understanding was that fairly large biases were a general problem until the corrections for mirror-temperature effects were developed (starting with baseline 2B10). Following this query we have looked briefly at monthly average biases for 2B06/2B07 but we found they were highly variable and sometimes in excess of 10 m/s so we don't think a comparison is useful.

Aeolus wind observations are filtered for an error estimate threshold of 8 m/s for
Rayleigh-clear and 4.5 m/s for Mie-cloudy winds. On which facts is this QC-criteria based? Did you try other values and how this affects the number of available data points and the determined random error of Aeolus wind observations?

The rejection thresholds of 4.5 m/s for Mie and 8 m/s Rayleigh errors were recommended by ESA/DISC (Aeolus Data Science and Innovation Cluster) (Reitebuch et al., 2019; Stoffelen et al., 2019; Rennie and Isaksen, 2020) for early period of Aeolus observations (laser FM-A) for the Aeolus CAL/VAL teams. For later period (laser FM-B) the limit for Mie was slightly changed for 5 m/s. These thresholds are chosen subjectively based on the compromise between the number of observations that pass quality control and the overall quality of the data set (Rennie and Isaksen, 2020). We followed the 1st recommendation and did not try other QCs. However, since we anyway needed to correct the application of our quality test for the radar winds, we have now redone all of the analysis with a rejection threshold of 5 m/s for Mie winds.

We will add to the text (P 6 l 150):

The rejection thresholds for Mie and Rayleigh winds were chosen as recommended by Rennie and Isaksen (2020). Those authors found appropriate thresholds subjectively based on a compromise between the number of observations that pass quality control and the overall quality of the data set.

Rennie, M. P. and Isaksen, L.: The NWP impact of Aeolus Level 2B winds at ECMWF, ECMWF technical memo, 864, doi: 10.21957/alift7mhr, 2020.

As horizontal collocation criteria a radius of 100 km around the radar wind profiler sites was chosen. The location of the two RWP sites close to the poles provide the opportunity to cover many Aeolus orbits at this latitudes where the satellite tracks are closer together. Did you try to increase the radius to a larger value (120 or 130 km) to see if this could improve the statistics by including more data points although the representativeness error would increase?

Following this suggestion, we tried increasing to 130 km radius. This has a negligible effect on the number of available comparison points for ESRAD (<2% more points). The biases are unchanged (within the 90% confidence intervals) but the standard-deviations and confidence intervals increase slightly (by up to 20% for monthly averages). For MARA, there is a bigger increase in available comparison points (14% for Rayleigh winds, 40% for Mie winds). Again, the biases do not change (within 90% confidence limits). For Rayleigh winds, standard deviations and confidence intervals are essentially unchanged, for Mie winds they are reduced by up to 30%. Since finding signs of uncorrected bias seems to us to be the most important aspect of this comparison, and a distance of 100 km is already rather far given the substantial topographic variation around the radar sites, we do not feel it is worthwhile to change our analysis to include more distant observations.

How does the inclusion of all Rayleigh-clear observations within the horizontal collocation radius affect the validation instead of using only the closest observations? This would be consistent with the approach which is applied for Mie-cloudy wind measurements and would provide more data points.

Rayleigh wind profiles are already averaged along 87 km of track. If we were to include further profiles with their mid-point within 100 km from the radar, this would mean observations up to 143.5 km away were included. We want to include points as close as possible. For Rayleigh winds there are enough of those using only the closest profile each pass. For Mie winds, there are too few comparison points if we take only the closest profiles.

Have the authors included the random errors of the RWP measurements as well as an estimate of the representativeness error in the determination of the Aeolus wind observation error? Otherwise, the determined random error of Aeolus would be a combination of the different errors.

In the paper we estimated standard deviation of Aeolus – radar wind difference which is combination of Aeolus observation error, representativeness error and RWP random error. We will reword the text of the paper to make it more clear.

We will change in 5. Summary … original p 16 l 329
*The random differences (4 - 7 m/s) are in most cases about what could be expected from the known level of random error for Aeolus winds (4-5 m/s for Rayleigh, 3 m/s for Mie) and spatial / temporal differences between radar and Aeolus measurements.*

The random differences are a combination of Aeolus observation error, representativeness error and radar wind random error. The values we observe (4 - 7 m s$^{-1}$) are in most cases about what could be expected from the known level of random error for Aeolus winds (4-5 m s$^{-1}$ for Rayleigh, 3 m s$^{-1}$ for Mie) and spatial / temporal differences between radar and Aeolus measurements.

An ERA5 model comparison which was performed in addition was mentioned several times in the manuscript. Did the authors consider to include this analysis in this work also in view of the possible imperfect bias correction of the ESRAD system?

We will add 2 new figures (see at the end of the reply) showing monthly average bias estimates for Aeolus vs radar, Aeolus vs ERA5, radar vs ERAS 5 and include references to this in the text.

P.1 L.20: Please include the data set time period and mention that reprocessed data was used in the analysis.

Will be included in the abstract and made clearer at the end of section 2.1.

P.3 L.65: Please include a short overview of the manuscript structure

We appreciate the suggestion but do not consider such overviews helpful.

P.3 L.76: "20 laser pulses": Until January 2019, 19 laser pulses were accumulated for one measurement. Afterwards it was only 18 pulses. See Weiler et al. 2020 or Lux et al. 2021 ("ALADIN laser frequency stability…")

20 will be changed to 18-20

P.3 L.79: Mie winds were horizontally averaged up to 14 km as of March 2019. Also horizontal averaging lengths smaller than 14 km are possible.

We will add (P.3 L.79)
(Also horizontal averaging lengths smaller than 14 km are possible).

P.4 L.14: How was the ESRAD bias correction done? Is this correction stable over all 6 months and the covered heights? From Belova et al. 2020, it seems not to be a constant offset over all heights.

We add more details in the text about this correction (see reply to RC 1). The final version of Belova et al. (2021) also includes more details. Note that it is not a 'bias', i.e. an offset, it is a systematic fractional underestimate, so higher wind speeds give higher differences.

P.5 Fig.2: What is plotted here? Are these the locations of the single wind observations in the L2B product? How does the number of Aeolus wind observations which are used in the comparison fit to the numbers (N points) from Table 2 and 3? For me it looks like there were much more Mie winds used than Rayleigh winds which is not represented by the numbers in the tables. Please clarify.

We will extend the figure text to clarify this as follows:
There are up to 14 (very sparsely populated) Mie-cloudy profiles within 100 km radius of the radar along each orbit track. These are combined to make a single profile for comparison with the single radar profile, as detailed in the text.

P.6 L.37: Is the QC-criteria based on an error estimate threshold applied before choosing the closest Rayleigh-clear wind observation or afterwards? Please clarify.

We can add :
The closest Rayleigh-clear profile is chosen before quality criteria are applied.

P.6 L.36: To better understand the statement "24 or 27 km" the authors could mention that the Aeolus range bins are following a digital elevation model (DEM) and within a radius of 100 km around the MARA site a strong change in topography from 0 to around 3000 m is covered.

We will add this.

P.6 L.40: Between mid October and December there was a so called Strateole range bin setting (RBS) with a maximum height of 17 km (17 km or 19-20 km for the MARA site) and an AMV setting with maximum height of 14 km for 2 weeks in between.

We will add this information.

P.6 L.42: The height resolution is only possible as a multiple of 250m. Please check if 1130 m should be changed to 1000 or 1250 m.

It should be 1000 m. We will correct this.

P.6 L.43: The AMV RBS with 14 km maximum height was in operation for 2 weeks.

We will correct this and make clear it applies at both sites.

P.6 L.44: See comment P.3 L.79

We will change 14 km to '14 km or shorter '

P.6 L.44: The Mie wind RBS are not the same as the Rayleigh RBS. For the sake of completeness, the Mie wind RBS could be added.
P.6 L.49: Why did the authors create a single profile out of all Mie wind observations and did not compare single Mie wind observations with RWP wind measurements? Are the obtained Mie wind profiles free of gaps? Since the Mie wind RBS are not the same as the Rayleigh RBS settings, additional errors could be introduced when averaging Mie winds to the vertical Rayleigh wind resolution.

Our understanding is that Mie wind measurements are only possible from cloud layers and these will be at well defined heights within range bins. Examination of the Mie-cloudy height profiles also shows that each profile usually includes valid wind in only zero or one or two height bins. Since our radar measurements are necessarily an average over an extended height interval, it is better to find enough Mie wind measurements to also average over height. The only way to do this is to collect together several Mie-cloudy profiles along the track and average to suitable height bins. Since the Mie-cloudy height bins change along the orbit track we decided just to use the same height bins as the closest Rayleigh profile. Since we have only a single radar profile available for each pass, we would not be able to calculate confidence intervals if we compared with multiple Mie profiles (correlated errors as all would be compared with the same radar profile).

Subsection 3.1: Could the authors please comment if blacklisted data was excluded?

There were no blacklisted data that affected our study. We checked it at
https://docs.google.com/document/d/1xCOqI3jxhSe8T9IpA5jb1BaRzqhyriQU3UpjZvY2Gz0/edit?usp=sharing

P.6 L.60: As already mentioned in comments above, the Mie and Rayleigh RBS are not the same. Are you also averaging to the Mie wind profiles?

See above.

P.8 L.16: To be consistent with the bias value in table 2 the value here should be changed to -1.9 m/s.

The text will be checked against the Tables and corrected.

Fig.4, 5, 10 and 11: Please increase the font size to the same value as used in Fig.2.

The figures will be replotted according to the comment.

Table 2 and 3: Can the authors please comment on the reason for the big difference in available data points for "Summer" and "Winter"? Are there less valid Aeolus or less valid RWP winds in winter?

Due to technical problems, the fca_4500 mode at MARA was not available 24 June-2 September, which reduces the available comparison points in the upper troposphere for winter at MARA. There are also mostly fewer valid Aeolus wind measurements for winter compared to summer (10 % fewer Mie winds in winter at ESRAD, 50% fewer at MARA, 30 % fewer Rayleigh wind measurements in winter at MARA, but 30 % more at ERSAD. A detailed analysis of the differences is beyond the scope of this paper, but we will add a comment on the fca_4500 problems to Table 1.

P.10 L.35: -2 m/s -> -1.9 m/s to be consistent with values in table 2

The text will be checked against the Tables and corrected.

P.13 Fig.10 b): Do you have any explanation for the strong negative bias for
descending tracks at around 9 km altitude? Why is the std difference between
ascending and descending orbits that large?

There is no obvious explanation for the bias at 9 km - this may just be an artefact of the small number on
comparison points.  Regarding the std difference - referring to Fig. 2b it is clear that the descending orbits are
further away from the radar than the ascending ones which suggests spatial variability as an explanation of the
larger std for descending orbits. We will add a comment on this to the text.

P.16 L.21: As mentioned above: up to 14 km

We will change (14 km) to (14 km or less)

New Figures to be included:

[Figure]

Figure X. Month-by-month mean values of biases in HLOS winds (solid lines) and 90% confidence limits (shaded areas) at MARA. Red for ascending tracks, blue for descending. a) Aeolus Rayleigh minus MARA, b) Aeolus Mie minus MARA, c) Aeolus Rayleigh minus ERA5, d) Aeolus Mie minus ERA5, e) MARA minus ERA5 at available Aeolus Rayleigh comparison times/heights, f) MARA minus ERA5 at available Aeolus Mie comparison times/heights.

[Figure]

Figure Y: Month-by-month mean values of biases in HLOS winds (solid lines) and 90% confidence limits (shaded areas) at ESRAD. Red for ascending tracks, blue for descending. a) Aeolus Rayleigh minus ESRAD, b) Aeolus Mie minus ESRAD, c) Aeolus Rayleigh minus ERA5, d) Aeolus Mie minus ERA5, e) ESRAD minus ERA5 at available Aeolus Rayleigh comparison times/heights, f) ESRAD minus ERA5 at available Aeolus Mie comparison times/heights.